# Hospital Outcomes in Patients Hospitalized for COVID-19 Pneumonia: The Effect of SARS-CoV-2 Vaccination and Vitamin D Status

**DOI:** 10.3390/nu15132976

**Published:** 2023-06-30

**Authors:** Martyna Sanecka, Modar Youssef, Mohammad Abdulsalam, Syed F. Raza, Abdul Qadeer, Julia Ioana, Alya Aldoresi, Syed I. Shah, Abdul Al Lawati, Joseph Feely, William P. Tormey, Eoghan O’Neill, Liam J. Cormican, Eoin P. Judge, Daniel M. A. McCartney, John L. Faul

**Affiliations:** 1School of Biological, Health & Sports Sciences, Technological University Dublin, D07 XT95 Dublin, Ireland; martynasanecka@gmail.com (M.S.); daniel.mccartney@tudublin.ie (D.M.A.M.); 2Department of Respiratory and Sleep Medicine, Connolly Hospital Dublin, D15 X40D Dublin, Ireland; 3Department of Biochemistry, Connolly Hospital Dublin, D15 X40D Dublin, Ireland; 4Department of Microbiology, Connolly Hospital Dublin, D15 X40D Dublin, Ireland; 5Department of Medicine, University College Dublin, D04 V1W8 Dublin, Ireland; 6Department of Medicine, Royal College of Surgeons in Ireland, D02 YN77 Dublin, Ireland

**Keywords:** COVID-19, SARS-CoV-2, vitamin D, 25-Hydroxyvitamin D (25(OH)D), hospitalization, mortality, vaccinated, unvaccinated

## Abstract

SARS-CoV-2 vaccination promises to improve outcomes for patients with COVID-19 pneumonia (most notably those with advanced age and at high risk for severe disease). Here, we examine serum 25-Hydroxyvitamin D (25(OH)D) status and outcomes in both old (>70 years) and young vaccinated (*n* = 80) and unvaccinated (*n* = 91) subjects, who were hospitalized due to COVID-19 pneumonia in a single center (Connolly Hospital Dublin). Outcomes included ICU admission and mortality. Serum 25(OH)D levels were categorized as D30 (<30 nmol/L), D40 (30–49.99 nmol/L) and D50 (≥50 nmol/L). In multivariate analyses, D30 was independently associated with ICU admission (OR: 6.87 (95% CI: 1.13–41.85) (*p* = 0.036)) and mortality (OR: 24.81 (95% CI: 1.57–392.1) (*p* = 0.023)) in unvaccinated patients, even after adjustment for major confounders including age, sex, obesity and pre-existing diabetes mellitus. While mortality was consistently higher in all categories of patients over 70 years of age, the highest observed mortality rate of 50%, seen in patients over 70 years with a low vitamin D state (D30), appeared to be almost completely corrected by either vaccination, or having a higher vitamin D state, i.e., mortality was 14% for vaccinated patients over 70 years with D30 and 16% for unvaccinated patients over 70 years with a 25(OH)D level greater than 30 nmol/L. We observe that high mortality from COVID-19 pneumonia occurs in older patients, especially those who are unvaccinated or have a low vitamin D state. Recent vaccination or having a high vitamin D status are both associated with reduced mortality, although these effects do not fully mitigate the mortality risk associated with advanced age.

## 1. Introduction

Vaccination promises to lower rates of infection and reduce disease mortality due to Severe Acute Respiratory Syndrome Coronavirus-2 (SARS-CoV-2) [1,2,3]. While evidence exists that vaccine efficacy may wane over time and with the emergence of virus mutations [1,2,3], it appears that vaccination does provide substantial protection against severe disease within the first six months of vaccination [1,2,3].

Chiu et al. have comprehensively reviewed the pathological immune responses which characterize severe SARS-CoV-2 infection and the mechanisms by which various vaccines including mRNA, DNA, protein subunit and inactivated viral vaccines mediate their immunogenicity [4]. These include the upregulation and IgM to IgG conversion of antibodies to the SARS-CoV-2 virus and its components (e.g., spike protein domains, the nucleocapsid), activation of toll-like receptors (TLRs), augmentation of type 1 Interferon response and enhanced helper T-cell and cytotoxic T-cell induction. Vitamin D (measured by serum 25-Hydroxyvitamin D (25(OH)D)) is thought to support these functions, including a smoother, accelerated transition to adaptive immunity via interferon-γ (IFN-γ) suppression, and an accelerated and more effective cytotoxic T-cell activity and B-cell antibody production via THαΒ cytokine T-cell responses to vaccination [4]. These effects suggest a confluent interface at which both vitamin D (1,25-dihydroxyvitamin D (1,25(OH)_2_D)) and vaccines act in concert with one another to amplify the immune response to the SARS-CoV-2 virus, but where in the absence of vaccine-induced immunity, vitamin D status may become a critical, independent determinant of effective immune response to the virus.

The exact effect of having a low vitamin D status on vaccine immune responses is unknown. While some studies have not identified enhanced antibody titers or IFN-γ response with higher vitamin D levels [5], other studies do observe differences in these immune responses according to vitamin D status. For example, in a prospective Italian study amongst 101 healthcare workers naïve for SARS-CoV-2 infection, significant correlations between the 25(OH)D concentration at baseline and the anti-spike antibody response and the overall neutralizing antibody titer at six months after the second vaccination dose were observed [6]. In a further prospective study amongst UK health workers, vaccine response (as measured by antibody production) was lower in those with low vitamin D status who received SARS-CoV-2 vaccination, again suggesting a weaker immunological response in individuals with lower serum 25-Hydroxyvitamin D (25(OH)D) measures [7]. Importantly perhaps, the latter study also revealed a weaker antibody response to SARS-CoV-2 vaccination in older adults, suggesting an independent effect of immuno-senescence on vaccine efficacy. This view is also supported in the literature where a large prospective study in Israel identified advanced age (as well as male sex and comorbidities such as hypertension and diabetes mellitus) to be a potent predictor of lower antibody response, particularly in relation to an initial mRNA vaccination dose [8].

It remains unknown whether being unvaccinated or having a low vitamin D state carry equal risk of severe disease, and also unclear is the extent to which advanced age influences this risk milieu. However, previous studies have indicated that the prevalence of severe COVID-19 disease is significantly higher in those who are vitamin D deficient or insufficient when compared to those with adequate vitamin D levels [9], while older age and absence of vaccination are established risk factors for poorer outcomes.

The current study aims to explore whether clinical outcomes in hospitalized COVID-19 pneumonia patients in Dublin, Ireland, vary according to vitamin D status, and whether any such effect differs between vaccinated and unvaccinated patients after adjustment for potential confounders, particularly age.

## 2. Materials and Methods

### 2.1. Study Design and Subject Cohort

This prospective cohort study enrolled 171 consecutive SARS-CoV-2 positive patients admitted to Connolly Hospital, Blanchardstown, Dublin 15 between June and December 2021. The Research Ethics Committee in the hospital approved this study and written informed consent was provided by participants enrolled prior to data analysis. “Vaccinated” participants received at least two doses of an EU-approved vaccine (*Pfizer* and/or *Jannsen*) within 6 months of hospital admission. Inclusion criteria were admission to Connolly Hospital, SARS-CoV-2 positivity confirmed on nasopharyngeal swab Polymerase Chain Reaction (PCR) analysis, proof of vaccination status, and serum 25(OH)D as measured on the day of admission. We excluded subjects with partial vaccination (receipt of only one dose of a COVID-19 vaccine or vaccination within 6 weeks of admission). We recorded demographics (age and sex), body mass index (BMI), smoking status, comorbidities (e.g., diabetes mellitus and hypertension), oxygen (O_2_) requirement for >24 h, mechanical ventilation, intensive care unit (ICU) admission and mortality.

### 2.2. Statistical Analysis

SPSS statistics software version 28.0 for Windows (IBM Corporation, Armonk, NY, USA) was used to conduct all statistical analyses. Histograms and normal Q-Q plots were evaluated visually, and Shapiro–Wilk testing undertaken to determine the normality of data distribution for all continuous variables. Categorical variables were expressed as absolute numbers (*n*) and relative frequencies (%), continuous variables were expressed as their mean and standard deviation (SD). BMI was divided into three categories: ideal weight (>18.5–24.99 kg/m^2^), overweight (25–29.99 kg/m^2^) and obese (≥30 kg/m^2^). Serum 25(OH)D measures were divided into three categories: D30 (<30 nmol/L), D40 (30–49.99 nmol/L) and D50 (≥50 nmol/L). Smoking status was divided into two categories, past or current smokers and non-smokers.

Dichotomous categorical variables were also generated for the following parameters: presence of comorbidities (diabetes mellitus, hypertension, etc.), requirement for O_2_ > 24 h, mechanical ventilation, ICU admission and mortality. Pearson’s Chi-square test was used to assess the association between all categorical patient variables and the clinical outcomes of interest. Yates’s continuity correction was used to determine statistical significance for all 2 × 2 cross-tabulations. For normally distributed continuous variables, one-tailed independent samples *t*-tests were used to compare means (age, 25(OH)D) between binary groups with differing clinical outcomes. These univariate analyses were used to determine if increased Coronavirus Disease 19 (COVID-19) disease severity (e.g., ICU admission, mortality) were significantly associated with factors such as older age or lower vitamin D levels in isolation. Binary logistic regression analyses were finally used to assess whether any of the factors highlighted on univariate analysis persisted as predictors of O_2_ requirement >24 h, mechanical ventilation, ICU admission or mortality after adjustment for confounding. For all statistical tests, a *p* value of less than <0.05 was used to define statistical significance.

## 3. Results

Baseline characteristics of all 171 subjects included in this study are presented according to vaccination status in Table 1. Vaccinated patients (average age of 69 years) were significantly older than unvaccinated patients (average age of 46 years) (*p* < 0.001). Comorbidities were commoner in the vaccinated group, with the exception of renal disease. The average number of comorbidities in vaccinated subjects was 2.68, compared to 0.68 in unvaccinated subjects (*p* < 0.001). Requirement for supplemental oxygen >24 h was similar between both groups (71% (*n* = 57) and 67% (*n* = 61) for vaccinated and unvaccinated groups, respectively). However, requirement for mechanical ventilation and ICU admission were higher in the unvaccinated group. Mortality was similar in both groups.

As seen in Table 2, no consistent association was observed between vitamin D status and requirement for O_2_ > 24 h, mechanical ventilation, ICU admission or mortality in the vaccinated group (*p* = 0.066, *p* = 0.694, *p* = 0.694 and *p* = 0.856, respectively). By contrast, in the unvaccinated group, stepwise reductions in the requirement for mechanical ventilation and ICU admission and in mortality were observed with increasing vitamin D status (e.g., three-to-fourfold differences in mechanical ventilation and ICU admission and eight-to-ninefold differences in mortality between the highest and lowest vitamin D categories), although these differences were not statistically significant (*p* = 0.21, *p* = 0.118 and *p* = 0.062, respectively). In sub-group analyses, the higher observed rates of mechanical ventilation (16.7% (*n* = 3) vs. 0% (*n* = 0)), ICU admission (27.8% (*n* = 5) vs. 0% (*n* = 0)) and mortality (22.2% (*n* = 4) vs. 7.1% (*n* = 1)) in those who were unvaccinated, and D30 (*n* = 18) compared to those who were vaccinated and D30 (*n* = 14) also failed to reach statistical significance (*p* = 0.321, *p* = 0.098 and *p* = 0.50, respectively).

Demographics (age and sex), smoking status, anthropometry (obesity), serum 25(OH)D and underlying disease according to O_2_ requirement, ICU admission and mortality are shown for vaccinated and unvaccinated patients in Table 3, Table 4 and Table 5.

Table 3 reveals that among the unvaccinated subjects, individuals with obesity (88.9%) and overweight (80.8%) were more likely to require extended supplemental O_2_ when compared to those who had a lower BMI (42.1%) (*p* < 0.001). Unvaccinated subjects who required prolonged supplemental oxygen had significantly lower levels of vitamin D (48.5 ± 20.9 nmol/L) compared to those who were vaccinated and required supplemental oxygen (64.4 ± 30.7 nmol/L) (*p* = 0.002, 95% CI 6.214–25.5). 

In unvaccinated patients, 25(OH)D was significantly lower in those admitted to ICU (41.2 ± 20.1 nmol/L) than in those who were not admitted (53.0 ± 23 nmol/L) (*p* = 0.033, 95% CI −0.83–24.51) (Table 4). Unvaccinated patients admitted to ICU were significantly younger than vaccinated patients admitted to ICU (48.9 ± 14.1 years vs. 69 ± 3.4 years; *p* = 0.029, 95% CI 2.27–37.863), although this may simply reflect the younger age profile of the unvaccinated group.

As presented in Table 5, in the vaccinated group only overweight and obesity predicted outcome, where they were associated with reduced risk of mortality (*p* = 0.025). By contrast, in the unvaccinated group, greater age (59 ± 17.8 years vs. 44.7 ± 14.0 years) and lower mean serum 25(OH)D (36.9 ± 15.1 nmol/L vs. 52.6 ± 23.1 nmol/L) were observed in unvaccinated patients who died than in those who survived (*p* = 0.005 and *p* = 0.049, respectively).

No statistically significant associations were observed between the requirement for mechanical ventilation and any of the explored predictive parameters, in either the vaccinated or unvaccinated groups.

The overall mortality rate was 10.5%. Mortality was higher in patients with advanced age (Figure 1). Increased mortality was also associated with a low vitamin D status, with similar death rates in young D30 subjects and older patients with D40 or D50. In the vaccinated cohort, in-hospital mortality was similar (11.3%) to that of the overall population; however, age appeared to be an important driver of increased mortality in these vaccinated patients rather than a low vitamin D state. Vaccinated patients aged over 70 years who were D30, D40 and D50 all had very similar mortality rates (~14–15%).

In unvaccinated patients, mortality was 9.9%. Mortality was higher with advanced patient age, but also increased as vitamin D status declined. Notwithstanding the low patient numbers in this category, mortality was three times higher in unvaccinated patients who were older and had low vitamin D status than in unvaccinated patients who had only one of these risk factors. Furthermore, mortality in unvaccinated patients was more than eight times higher in those who were older and had low vitamin D status than in unvaccinated patients who had neither of these risk factors. Unvaccinated subjects who died were younger (59 ± 17.2 vs. 76.8 ± 7.3 years years) and had significantly lower levels of vitamin D (36.9 ± 15.1 nmol/L vs. 62.8 ± 26.9 nmol/L) than those who were vaccinated and died (*p* = 0.011 (CI 4.59–30.95) and *p* = 0.023 (CI 4.09–47.72), respectively).

For vaccinated patients in both age groups, there was no protective effect associated with a serum 25(OH)D level greater than 30 nmol/L. For unvaccinated patients in both age groups, however, mortality was increased threefold in patients with serum 25(OH)D less than 30 nmol/L.

In summary, advanced age increased mortality in vaccinated patients irrespective of vitamin D status; in unvaccinated patients, however, both advanced age and low vitamin D status increased mortality, and these effects were additive.

Binary logistic regression analyses were performed to assess the association between a set of predictor variables and the likelihood of ICU admission, mortality, extended supplemental O_2_ requirement and mechanical ventilation. Table 6 and Table 7 explore the relationship between ICU admission and mortality and six independent variables (age category, sex, BMI, vitamin D category, smoking status and disease score category) in unvaccinated patients.

As presented in Table 6, after adjustment for major confounders, unvaccinated patients who were D30 (25(OH)D < 30 nmol/L) had a significantly greater likelihood of requiring ICU admission than unvaccinated patients who were D50 (25(OH)D > 50 nmol/L) (OR: 6.87 (95% CI: 1.13–41.85) (*p* = 0.036).

When a similar multivariate model was applied to mortality in unvaccinated patients (Table 7), those aged between 60–79 years were significantly more likely to die than those aged 17–39 years (OR: 66.4 (95% CI: 1.98–2222.5) (*p* = 0.019). Similarly, unvaccinated patients who were vitamin D deficient (25(OH)D < 30 nmol/L) had a significantly greater likelihood of mortality than those with 25(OH)D > 50 nmol/L (OR: 24.807 (95% CI: 1.57–392.1) (*p* = 0.023).

Similar binary logistic regression analyses were performed for extended supplemental O_2_ requirement and mechanical ventilation in the unvaccinated cohort, but no statistically significant associations between the predictor variables and these clinical outcomes were apparent. Similarly, binary logistic regression analyses evaluating the independent associations of these variables with the four clinical outcomes in vaccinated patients yielded no significant findings.

## 4. Discussion

This prospective cohort study highlights five important differences between vaccinated and unvaccinated patients hospitalized for COVID-19 pneumonia. First, vaccinated patients were on average more than 20 years older than unvaccinated patients. Second, vaccinated subjects had more co-morbidity. Third, the average vitamin D levels were significantly lower in unvaccinated patients who died (mean 36.9 nmol/L) compared to vaccinated subjects who died (62.8 nmol/L). Fourth, advanced age (greater than 70 years) was associated with higher mortality when comparing vaccinated and unvaccinated patients and when comparing those with low and high vitamin D status. Finally, in unvaccinated patients, low vitamin D levels (D30) were associated with a nearly sevenfold increased risk of ICU admission and with an almost 25-fold increased risk of mortality even after adjustment for major confounders such as age, sex, obesity and pre-existing disease—trends which were not apparent in vaccinated patients.

These data confirm recent reports of significantly higher age (73 years versus 67 years) and lower incidence of pneumonia (69% versus 93%) in vaccinated subjects, who were admitted to hospital with breakthrough infections when compared to unvaccinated subjects [10]. This also appears to be evident on a population basis. In a study of COVID-19 mortality in Los Angeles County, USA, in the summer of 2021, for example, vaccinated subjects who died had significant co-morbidity, (including HIV and cancer) and their median age was more than 10 years older (74 years), compared to a median of 63 years amongst unvaccinated subjects who died [11], supporting the idea that advanced age carries significant risk for mortality in vaccinated subjects. In our study, the average age of vaccinated patients who died was 77 years, compared with 59 years for unvaccinated subjects. Amongst unvaccinated patients, however, our data show similar mortality rates in younger adults who have a low vitamin D state as in older age groups with higher vitamin D serum measures, suggesting that the presumptive reduction in mortality risk associated with younger age is lost in unvaccinated younger patients who have lower vitamin D.

Our findings match our previous findings and other studies which demonstrate a relation between low 25(OH)D levels upon hospital admission and poorer COVID-19 disease outcomes [12,13,14]. Here, we included comparisons of serum 25(OH)D levels on the day of admission between vaccinated (against SARS-CoV-2) and unvaccinated patients hospitalized for COVID-19 pneumonia. While we found a high prevalence of D30 (17.5% and 19.8%, respectively), and D40 (22.5% and 37.4%, respectively) in both vaccinated and unvaccinated patients, unvaccinated COVID-19 pneumonia patients with 25(OH)D < 30 nmol/L on admission were nearly 25 times more likely to die even after adjustment for major confounders. A serum 25(OH)D < 30 nmol/L on admission in unvaccinated COVID-19 patients was also significantly and independently associated with ICU admission (OR: 6.87 (95% CI: 1.13–41.85) (*p* = 0.036). By contrast, no significant association between vitamin D status and mortality, ICU admission or mechanical ventilation was seen in the vaccinated group, suggesting vaccination provides protection against severe disease, regardless of vitamin D status. An alternative explanation is that the vaccinated group of patients are at risk of severe disease on account of their advanced age (on average over 20 years older) and increased number of co-morbidities, and these effects overshadow the putative immuno-protective effects of replete vitamin D status [10].

### Strengths and Limitations

This study has several strengths: First, this is a single-center prospective study, avoiding the possible confounding effects of recruiting subjects from a wide variety of geographic locales or healthcare facilities. Second, we were able to recruit patients when SARS-CoV-2 vaccination was first becoming available. Third, our population of vaccinated and unvaccinated patients were all Caucasian, avoiding the potential confounding effect of ethnicity on vitamin D status. A weakness of this study is that we did not manage to recruit dialysis patients or solid organ allograft recipients; therefore, we cannot determine whether the effects that we have described are applicable to these populations. In addition, our sample size may lack the power to demonstrate an association between vitamin D status and mechanical ventilation because this occurred in such a low number of patients (a type 2 error). We did not confirm the effects of obesity on COVID-19 disease severity. Here, body weight (measured by BMI) was inversely associated with mortality in vaccinated patients (*p* = 0.025) and positively with O_2_ requirement in unvaccinated patients (*p* < 0.001) on univariate analysis; however, these associations did not persist on multivariate analysis, suggesting that these effects might have been attributable to the presence of confounders associated with overweight and obesity (i.e., collinearity).

The findings of the current study support our previous findings, i.e., that a low vitamin D status may increase disease severity, at least amongst unvaccinated patients [14]. The current study includes subjects with very low serum vitamin D measures (25(OH)D < 30 nmol/L), indicating that mortality risk rises considerably in unvaccinated patients in this D30 group, and perhaps more modestly among those in the D40 group when compared with their more vitamin D replete peers. A number of reviews have discussed the possibility that a low vitamin D state is associated with worse outcome in patients with COVID-19 [15,16,17,18] A systematic review and meta-analysis, which pooled data from one population study and seven clinical studies (two independent datasets) reported a significant inverse correlation between patient vitamin D levels and SARS-CoV-2 mortality (*r =* −0.3989, *p* = 0.02) [15]. In these studies, vitamin D levels were notably collected before infection or within the first day of hospital admission, reducing the likelihood that these findings were attributable to reverse causality. Further meta-analyses of intervention studies indicate that the likelihood of severe COVID-19 disease and of COVID-19 mortality is considerably lower in those receiving vitamin D supplements [16,17]. One meta-analysis comprising cohort studies, RCTs and multivariate-adjusted studies found no correlation between low serum vitamin D levels and negative clinical outcomes in COVID-19 patients, based on the findings that serum 25(OH)D levels <50 nmol/L or <75 nmol/L were not associated with in-hospital mortality (OR 2.18, 95% CI: 0.91–5.26 and OR 3.07, 95% CI: 0.64–14.78, respectively) [18]. These findings further support the idea that the threshold for substantial increase in COVID-19 mortality is likely to lie closer to 30 nmol/L than 50 nmol/L. In support of the effect of vitamin D status on COVID-19 immunity, recent data have demonstrated enhanced COVID-19 vaccine response in patients who are vitamin D replete [6,7], while earlier work has also elucidated the mechanisms by which vitamin D likely mediates these positive effects on COVID-19 immunity [19]. 

The current data suggest that D30 is associated with increased risk of mortality and ICU admission amongst unvaccinated subjects. This substantially increased risk was seen in both younger (<70 years) and older (>70 years) adults. Meta-analyses of prospective observational studies have previously demonstrated an association between low vitamin D status and more severe COVID-19 disease including the need for ICU admission [20,21]. Further systematic reviews of intervention studies have reported significantly lower likelihood of severe COVID-19 disease and mortality with vitamin D supplementation [16,20,22,23], including reduced risk of ICU admission in patients receiving vitamin D supplements (OR 0.35, 95% CI: 0.28–0.44) [16]. A German prospective study which reported on 185 consecutive SARS-CoV-2 positive patients found that 62% of those with D30 required high-flow oxygen or mechanical ventilation compared with 27% of those who were D50 (*p* = 0.004) [9]. Further studies have also identified an association between low vitamin D status and increased risk of mechanical ventilation [9,12]. It is noteworthy, though, that many of these studies precede the availability of vaccines. In the current study, we found no association between vitamin D status and requirement for mechanical ventilation on either univariate or multivariate analyses, most likely due to the small number of patients overall who received mechanical ventilation (*n* = 14, 8.2%).

Low vitamin D status is highly prevalent in Ireland and other northern countries [24]. The exact role of vitamin D in COVID-19 immunity is unknown, but vitamin D appears to affect immune responses to viral respiratory infection [25]. During SARS-CoV-2 infection, vitamin D downregulates the expression of many of the pro-inflammatory cytokines which can ultimately lead to multi-organ failure both directly and indirectly. These include interleukin-1 (IL-1), IL-6, IL-18, IL-10, interferon gamma (IFN-γ) and tumor necrosis factor-alpha (TNF-α) secreted by T helper type 1 (T_H_1) and other cells during the inflammatory process of COVID-19. CD4^+^ T cells of patients with severe COVID-19 appear to be T_H_1-skewed and show de-repression of genes that are downregulated by vitamin D, from either lack of substrate (a low vitamin D state) or abnormal regulation of this system [26,27].

It is clear that host-related factors mediate much of the variability in severity of SARS-CoV-2 infection, thus changing survival and clinical outcomes in COVID-19 [28]. While the excess risk associated with older age and pre-existing comorbidities is now well established, the metabolic and immunological roles described above suggest that vitamin D deficiency may be a further important host-related risk factor for severe COVID-19 disease [29,30,31]. The overlap between the risk factors for vitamin D deficiency and those associated with severe COVID-19 disease has complicated the issue of causality. In this regard, however, low vitamin D status has persisted as a risk factor for severe COVID-19 disease outcomes in several studies which have adjusted for these confounders [14,30], including this one. Furthermore, recent meta-analyses have shown that supplementation with vitamin D may help to reduce COVID-19 disease severity and mortality [16,17,20,22,23], helping to consolidate a causal role for low vitamin D status in severe COVID-19 disease [31,32].

## 5. Conclusions

This paper confirms that a low vitamin D status is associated with adverse COVID-19 outcomes in unvaccinated individuals, supporting the idea that vitamin D has important immune-related effects [33], in particular against severe SARS-CoV-2 infection. Together with the current literature, this supports the idea that serum levels of 25(OH)D above a minimum target threshold of 50 nmol/L at all times of year might provide important protection against severe COVID-19 disease [31], especially for younger unvaccinated patients without other significant co-morbidity. If vaccine-induced immunity wanes over time, or if new variants evade vaccine associated immunity, vitamin D supplementation may prove an even more important intervention in mitigating these risks.

## Figures and Tables

**Figure 1 nutrients-15-02976-f001:**
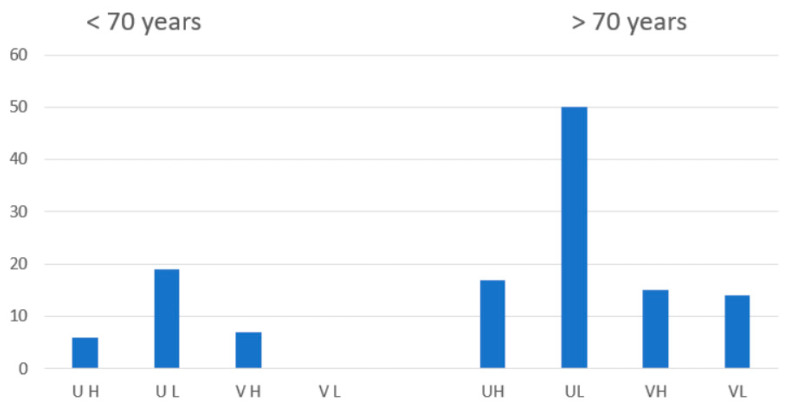
Bar chart exploring mortality (% on the *y*-axis) of vaccinated (V) and unvaccinated (U) patients according to their age (<70 or >70 years) and vitamin D status (serum 25(OH)D < 30 nmol/L (low (L) or >30 nmol/L (high (H)). UH denotes unvaccinated subjects with a “high” vitamin D state; UL denotes unvaccinated subjects with a “low” vitamin D state; VH denotes vaccinated subjects with a “high” vitamin D state; VL denotes vaccinated subjects with a “low” vitamin D state.

**Table 1 nutrients-15-02976-t001:** Socio-demographic, anthropometric and clinical status of COVID-19 patients.

Characteristics (*n* = 171)	Vaccinated (*n* = 80)	Unvaccinated (*n* = 91)	*p* Value
Age (years)			
Mean ± (SD)	69 (16)	46 (15)	<0.001
Sex			
Female	40 (50)	41 (45.1)	0.622
BMI			
Overweight (25–29.99 kg/m^2^)	16 (20)	26 (28.6)	
Obese (>30 kg/m^2^)	26 (32.5)	27 (29.7)	0.428
Smoking Status			
Past/Current Smoker	10 (12.5)	8 (8.8)	0.59
Comorbidity			
Respiratory Disease	24 (30)	2 (2.2)	<0.001
Diabetes Mellitus	18 (22.5)	5 (5.5)	0.002
Hypertension	43 (53.8)	16 (17.6)	<0.001
Hyperlipidaemia	25 (31.3)	4 (4.4)	<0.001
Renal Disease	9 (11.3)	4 (4.4)	0.162
Malignancy	12 (15)	3 (3.3)	0.015
Ischemic Heart Disease	24 (30)	4 (4.4)	<0.001
25(OH)D Concentration (nmol/L)			
D30, D40 (<50 nmol/L)	32 (40)	42 (57.1)	0.059
D50 (≥50 nmol/L)	48 (60)	39 (42.9)	
O_2_ Requirement			
Yes	57 (71.3)	61 (67)	0.688
Mechanical Ventilation Requirement			
Yes	3 (3.8)	11 (12.1)	0.088
ICU Admission	3 (3.8)	15 (16.5)	0.014
Survival to Discharge	71 (88.8)	82 (90.1)	0.969

Categorical variables expressed as their total number *n* and percentage (%), while age is expressed as mean and standard deviation. Body mass index (BMI), intensive care unit (ICU), oxygen (O_2_) serum 25-hydroxyvitamin D (25(OH)D, standard deviation (SD)).

**Table 2 nutrients-15-02976-t002:** Clinical outcomes of patients stratified according to vitamin D status on hospital admission.

Outcomes	Vaccinated *n*= 80					Unvaccinated *n* = 91				
	Total *n* = 80	D30 *n* = 14	D40 *n* = 18	D50 *n* = 48	*p*	Total *n* = 91	D30 *n* = 18	D40 *n* = 34	D50 *n*= 39	*p*
O_2_ Requirement Required	57 (71.3)	10 (71.4)	9 (50)	38 (79.2)	0.066	60 (65.9)	13 (72.2)	24 (73.5)	23 (59)	0.365
Mechanical Ventilation Requirement Required	3 (3.8)	0 (0.0)	1 (5.6)	2 (4.2)	0.694	11 (12.1)	3 (16.7)	6 (17.6)	2 (5.1)	0.21
ICU Admission Admitted	3 (3.8)	0 (0.0)	1 (5.6)	2 (4.2)	0.694	15 (16.5)	5 (27.8)	7 (20.6)	3 (7.7)	0.118
Mortality Deceased	9 (11.3)	1 (7.1)	2 (11.1)	6 (12.5)	0.856	9 (9.9)	4 (22.2)	4 (11.8)	1 (2.6)	0.062

D30 denotes a serum 25(OH)D less than 30 nmol/L, D40 denotes a serum 25(OH)D greater than 30 nmol/L, but less than 49.9 nmol/L; D50 denotes a serum 25(OH)D greater than 50 nmol/L. All variables are expressed as their total number *n* and percentages (%). Intensive care unit (ICU), oxygen (O_2_).

**Table 3 nutrients-15-02976-t003:** Demographic, lifestyle (smoking), anthropometric and underlying health characteristics and vitamin D measures stratified according to supplemental O_2_ requirement in vaccinated and unvaccinated patients.

Variable	Vaccinated		Unvaccinated	
	O_2_ Requirement *n* = 57	*p* (95% CI)	O_2_ Requirement *n* = 61	*p* (95% CI)
Age (years) (mean ± SD) Required supplemental O_2_ Did not require supplemental O_2_	70.9 (14.7) 64.9 (18)	0.061 (−13.76–1.64)	47.41 (14.84) 43.43 (14.76)	0.116 (−10.54–2.587)
Sex Female Male	26 (65.0) 31 (77.5)	0.323	24 (58.6) 37 (0.74)	0.181
Past/Current Smoker	7 (70.0)	1.000	5 (62.5)	1.000
BMI Category Ideal (18.5–24.99 kg/m^2^) Overweight (25–29.99 kg/m^2^) Obese (≥30 kg/m^2^)	27 (71.1) 9 (56.25) 21 (80.77)	0.234	16 (42.1) 21 (80.8) 24 (88.9)	<0.001
25(OH)D Concentration (nmol/L) (mean ± SD) Required supplemental O_2_ Did not require supplemental O_2_	64.37 (30.66) 55.19 (34.65)	0.123 (−24.84–6.47)	48.52 (20.87) 56.26 (26.10)	0.065 (−2.32–17.80)
Individual Comorbidities *n* (%) Respiratory Disease Diabetes Mellitus Hypertension Hyperlipidaemia Renal Disease Malignancy Ischaemic Heart Disease	17 (70.1) 14 (77.8) 29 (67.4) 20 (80.0) 9 (100.0) 9 (75.0) 19 (79.2)	1.000 0.690 0.573 0.368 0.103 1.000 0.450	2 (100) 5 (100) 13 (81.3) 3 (75.0) 3 (75.0) 2 (66.7) 2 (50.0)	0.809 0.261 0.299 1.000 1.000 1.000 0.844
Cumulative Comorbidity Score *n* (%) 0–2 3–4 5–6	20 (62.5) 25 (73.5) 12 (85.7)	0.258	57 (66.3) 4 (80.0) 0 (0.0)	0.885

Continuous variables expressed as mean and standard deviation, Categorical variables expressed as their total number *n* and percentages (%). Oxygen (O_2_), body mass index (BMI), serum 25-hydroxyvitamin D (25(OH)D), 95% confidence interval (95% CI), standard deviation (SD).

**Table 4 nutrients-15-02976-t004:** Demographic, lifestyle (smoking), anthropometric and underlying health characteristics and vitamin D measures stratified according to ICU admission in vaccinated and unvaccinated patients.

Variable	Vaccinated		Unvaccinated	
	ICU Admission *n* = 3	*p* (95% CI)	ICU Admission *n* = 15	*p* (95% CI)
Age (years) (mean ± SD) Admitted to ICU Not admitted to ICU	69.0 (3.46) 69.19 (16.09)	0.492 (−18.3–18.82)	48.93 (14.13) 45.54 (15.02)	0.211 (−11.747–4.99)
Sex Female Male	0 (0.0) 3 (7.5)	0.239	7 (17.1) 8 (16.0)	1.000
Past/Current Smoker	0 (0.0)	1.000	0(0.0)	0.414
BMI Category Ideal (18.5–24.99 kg/m^2^) Overweight (25–29.99 kg/m^2^) Obese (≥30 kg/m^2^)	2 (5.3) 1 (6.3) 0 (0.0)	0.465	4 (10.5) 7 (26.9) 4 (14.8)	0.213
25(OH)D Concentration (nmol/L) (mean ± SD) Admitted to ICU Not admitted to ICU	63.82 (16.15) 61.66 (32.42)	0.455 (−39.78–35.46)	41.18 (20.01) 53.02 (23.01)	0.033 (−0.83–24.51)
Individual Comorbidities *n* (%) Respiratory Disease Diabetes Mellitus Hypertension Hyperlipidaemia Renal Disease Malignancy Ischaemic Heart Disease	0 (0.0) 2 (11.1) 1 (2.3) 2 (8.0) 1 (11.1) 1 (8.3) 0 (0.0)	0.607 0.245 0.894 0.475 0.762 0.934 0.607	0 (0.0) 2 (40.0) 3 (18.8) 0 (0.0) 1 (25.0) 0 (0.0) 0 (0.0)	1.000 0.402 1.000 0.826 1.000 1.000 0.826
Cumulative Comorbidity Score *n* (%) 0–2 3–4 5–6	1 (3.2) 1 (2.9) 1 (7.1)	0.762	14 (16.3) 1 (20.0) 0 (0.0)	1.000

Continuous variables expressed as mean and standard deviation, Categorical variables expressed as their total number *n* and percentages (%). Intensive care unit (ICU), body mass index (BMI), serum 25-hydroxyvitamin D (25(OH)D), 95% confidence interval (95% CI), standard deviation (SD).

**Table 5 nutrients-15-02976-t005:** Demographic, lifestyle (smoking), anthropometric and underlying health characteristics and vitamin D measures stratified according to mortality in vaccinated and unvaccinated patients.

Variable	Vaccinated		Unvaccinated	
	Mortality *n* = 9	*p* (95% CI)	Mortality *n* = 9	*p* (95% CI)
Age (years) (mean ± SD) Died Survived	76.8 (7.3) 68.2 (16.3)	0.126 (−2.45–19.58)	59 (17.2) 44.7 (14.0)	0.005 (4.34–24.29)
Sex Female Male	3 (7.5) 6 (15)	0.479	5 (12.2) 4 (8)	0.753
Past/Current Smoker	0 (0.0)	0.504	0 (0.0)	0.718
BMI Category Ideal (18.5–24.99 kg/m^2^) Overweight (25–29.99 kg/m^2^) Obese (≥30 kg/m^2^)	8 (21.1) 1 (6.3) 0 (0.0)	0.025	3 (7.9) 3 (11.5) 3 (11.1)	0.863
25(OH)D Concentration (nmol/L) (mean ± SD) Died Survived	62.8 (26.9) 61.6 (32.6)	0.919 (−21.46–23.78)	36.9 (15.1) 52.6 (23.1)	0.049 (−31.47–−0.071)
Individual Comorbidities *n* (%) Respiratory Disease Diabetes Mellitus Hypertension Hyperlipidaemia Renal Disease Malignancy Ischaemic Heart Disease	1 (4.2) 2 (11.1) 4 (9.3) 3 (12) 2 (22.2) 1 (8.3) 2 (8.3)	0.354 1.000 0.811 1.000 0.585 1 0.877	0 (0.0) 1 (20) 1 (6.3) 0 (0.0) 0 (0.0) 0 (0.0) 0 (0.0)	1.000 0.993 0.939 1.000 1.000 1 1
Cumulative Comorbidity Score *n* (%) 0–2 3–4 5–6	3 (9.4) 4 (11.8) 2 (14.3)	0.882	9 (10.5) 0 (0.0) 0 (0.0)	1

Continuous variables expressed as mean and standard deviation, Categorical variables expressed as their total number *n* and percentages (%). Body mass index (BMI), serum 25-hydroxyvitamin D (25(OH)D), 95% confidence interval (95% CI), standard deviation (SD).

**Table 6 nutrients-15-02976-t006:** Binary logistic regression analysis of putative factors associated with ICU admission in unvaccinated patients.

Variable	Beta Coefficient	SE	OR (95% CI)	*p* Value
Age Category				
40–59	1.719	0.76	1.758 (0.396–7.798)	0.458
60–79	18.277	1.228	7.68 (0.692–85.305	0.097
80–99	19.47	23,061.919	0 (0)	0.999
Sex				
Male	−0.358	0.722	0.699 (0.17–2.876)	0.62
BMI				
Overweight (25–29.99 kg/m^2^)	0.639	0.813	1.894 (0.385–0.151)	0.432
Obese (≥30 kg/m^2^)	−0.229	0.848	0.795 (0.151–4.192)	0.787
Vitamin D Category				
Insufficient (D40)	1.139	0.822	3.123 (0.623–15.645)	0.166
Deficient (D30)	1.927	0.919	6.868 (1.134–41.582)	0.036
Past/Current Smoker	−20.335	12,727.432	0	0.999
Disease Score Category				
3–4	−0.492	1.379	0.611 (0.41–9.126)	0.721

Reference categories: age < 40, female sex, BMI < 25 kg/m^2^, vitamin D sufficiency (25(OH)D ≥ 50 nmol/L), never smoking, 0–2 comorbidities, odds ratio (OR), standard error (SE), 95% confidence interval (95% CI), body mass index (BMI).

**Table 7 nutrients-15-02976-t007:** Binary logistic regression analysis of putative factors associated with mortality in unvaccinated patients.

Variable	Beta Co-Efficient	SE	OR (95% CI)	*p* Value
Age Category				
40–59	2.106	1.275	8.217 (0.676–99.918)	0.098
60–79	4.196	1.791	66.392 (1.983–2222.539)	0.019
80–99	2.058	1.804	7.831 (0.228–268.821)	0.254
Sex				
Male	0.311	1.028	1.365 (0.182–10.23)	0.762
BMI				
Overweight (25–29.99 kg/m^2^)	−0.383	1.188	0.682 (0.066–6.998)	0.747
Obese (≥30 kg/m^2^)	−0.35	0.995	0.705 (0.1–4.949)	0.725
Vitamin D Category				
Insufficient (D40)	1.116	1.226	3.052 (0.276–33.728)	0.363
Deficient (D30)	3.211	1.408	24.807 (1.57–392.062)	0.023
Past/Current Smoker	−20.952	11,458.453	0	0.999
Disease Score Category				
3–4	−20.202	17,395.914	0	0.999

Reference categories: age < 40, female sex, BMI < 25 kg/m^2^, vitamin D sufficiency (25(OH)D ≥ 50 nmol/L), never smoking, 0–2 comorbidities, odds ratio (OR), standard error (SE), 95% confidence interval (95% CI), body mass index (BMI).

## Data Availability

The data presented in this study are available on request from the corresponding author. The data are not publicly available due to privacy restrictions.

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
