# Peer review of "Hospital Outcomes in Patients Hospitalized for COVID-19 Pneumonia: The Effect of SARS-CoV-2 Vaccination and Vitamin D Status"

_nutrients, 2023, doi:10.3390/nu15132976_

Round 1

Reviewer 1 Report

The authors present a valuable prospective cohort study linking low vitamin D status to higher rates of mortality and ICU admittance for unvaccinated SARS-CoV-2 patients.

Major Comments:
All original data discussed should reference figures or tables in the results section. Some of the data appears only to be available in paragraph form. For example: "Among the unvaccinated subjects, obese (88.9%) and overweight (80.8%) individuals 117 were more likely to require extended supplemental O2 when compared to those who had 118 a lower BMI (42.1%) (p=<0.001)." No table or figure is referenced and the breakdown of supplemental O2 requirement by vaccination and overweight/obese status does not appear to be available.

Relationships that are not found to be significant should not be reported as "trends" in the text. That defeats the purpose of significance testing. However, the authors can use significance tests that specifically address their questions. For instance, the authors state: "In unvaccinated patients, 25(OH)D was lower in those admitted to ICU (41.2 ± 20.1 123 nmol/L) than in those who were not admitted (53.0 ± 23 nmol/L), although this trend did 124 not reach statistical significance (p=0.067)." If the original question is, "Is vitamin D status lower in unvaccinated patients admitted to the ICU?", then a one-tailed t-test would be appropriate, resulting a p-value that is half of the two-tailed p-value.

Proteomic studies reveal vitamin D status to have diverse impacts on protein expression, including those associated with lung development. These impacts are reviewed in Jeong and Vacanti, Nutrition and Metabolism 2020. This should be cited as a potential mechanism predisposing unvaccinated patients to adverse effects.

Author Response

Cover letter and response to reviewers:

Thank you for asking us to re-submit this paper and thank you for including the reviewers’ comments.  We have modified the paper accordingly and we believe that the paper is now significantly strengthened.

Comments and Suggestions for Authors

The authors present a valuable prospective cohort study linking low vitamin D status to higher rates of mortality and ICU admittance for unvaccinated SARS-CoV-2 patients.

Major Comments:

All original data discussed should reference figures or tables in the results section. Some of the data appears only to be available in paragraph form. For example: "Among the unvaccinated subjects, obese (88.9%) and overweight (80.8%) individuals 117 were more likely to require extended supplemental O2 when compared to those who had 118 a lower BMI (42.1%) (p=<0.001)." No table or figure is referenced and the breakdown of supplemental O2 requirement by vaccination and overweight/obese status does not appear to be available.

A new table 2 has been inserted for clarity. We now describe that the above data are contained in Table 3a. A further breakdown of co-morbidities is contained in Table 3b. In other areas, where a table is unnecessary, we state “data not shown”.

Relationships that are not found to be significant should not be reported as "trends" in the text. That defeats the purpose of significance testing. However, the authors can use significance tests that specifically address their questions. For instance, the authors state: "In unvaccinated patients, 25(OH)D was lower in those admitted to ICU (41.2 ± 20.1 123 nmol/L) than in those who were not admitted (53.0 ± 23 nmol/L), although this trend did 124 not reach statistical significance (p=0.067)." If the original question is, "Is vitamin D status lower in unvaccinated patients admitted to the ICU?", then a one-tailed t-test would be appropriate, resulting a p-value that is half of the two-tailed p-value.

We have changed the description of the statistical testing and the analysis to a one-tailed t -test.

Proteomic studies reveal vitamin D status to have diverse impacts on protein expression, including those associated with lung development. These impacts are reviewed in Jeong and Vacanti, Nutrition and Metabolism 2020. This should be cited as a potential mechanism predisposing unvaccinated patients to adverse effects.

This reference has been included as reference 33 and is referenced in the body of the text.

Reviewer 2 Report

The authors compared vaccinated and unvaccinated patients with Covid-19 pneumonia aiming to determine whether a low vitamin D status affects their hospital course. This manuscript was well-designed and written but some issues should be addressed before publication. Please see below:

i)                    The introduction section should be improved. The authors should present an overview of the evaluation of Vitamin D and vaccination in patients with Covid-19 pneumonia. There are many papers in the literature;

ii)                   The caption of Figure 1: the abbreviations (e.g. U, V, UL…) should be improved. 

Author Response

1. The introduction section should be improved. The authors should present an overview of the evaluation of Vitamin D and vaccination in patients with Covid-19 pneumonia. There are many papers in the literature.

The introduction has been extensively lengthened with a more detailed description of the current understanding of vitamin D in Covid-19 and the role of vaccination.

2. The caption of Figure 1: the abbreviations (e.g. U, V, UL…) should be improved.

We thank the reviewer for this comment. The caption has been rewritten and each abbreviation is now clarified to better explain the labels in the figure.

Reviewer 3 Report

The manuscript compares vaccinated and unvaccinated patients with Covid-19 pneumonia to determine whether a low vitamin D status affects their hospital course.

Please address the following suggestions:

Please introduce the definition of every abbreviation the first time you use it in the manuscript (i.e. ICU, 25(OH)D, etc.).

I advise the authors to expand the Introduction section by using relevant information on the topic they studied.

Please add a clear aim of the study at the end of the Introduction section.

I believe that a Conclusions section and a section for the Limitations of the study should be inserted in the manuscript.

Minor editing of English language required

Author Response

Please introduce the definition of every abbreviation the first time you use it in the manuscript (i.e. ICU, 25(OH)D, etc.). This has been done.

I advise the authors to expand the Introduction section by using relevant information on the topic they studied. The introduction has been extensively lengthened with a more detailed description of the current understanding of vitamin D in Covid-19 and the role of vaccination.

Please add a clear aim of the study at the end of the Introduction section. The introduction has been extensively lengthened with a more detailed description of the current understanding of vitamin D in Covid-19 and the role of vaccination. A clear aim is now included at the end of the introduction.

I believe that a Conclusions section and a section for the Limitations of the study should be inserted in the manuscript.

A section for conclusions and a section on “strengths and limitations” have been included in the discussion.

Round 2

Reviewer 3 Report

The manuscript compares vaccinated and unvaccinated patients with Covid-19 pneumonia to determine whether a low vitamin D status affects their hospital course.

Please address the following suggestions:

Please introduce the definition of EVERY abbreviation the first time you use it in the manuscript.

Please refrain from using citations in the Conclusion section.

I advise the authors to include the section of Strengths and limitations (only the recently added text) right before the Conclusions section. I do not consider that this section is part of the Discussions section, thus it does not require citations.

Minor editing of English language required